# Age differences in the conceptualization and experience of curiosity: A qualitative study

**Michelle E. Hirsch**[1¤*], **Tarnpreet Virk**[1], **Christina Chang**[1,2], **Buddhika Bellana**[1,2,3], **Andrée-Ann Cyr**[1,2]

**1** Department of Psychology, York University, Toronto, Ontario, Canada, **2** Department of Psychology, York University Glendon Campus, Toronto, Ontario, Canada, **3** Rotman Research Institute, Baycrest Academy for Research & Education, Toronto, Ontario, Canada

¤ 4700 Keele St, North York, Ontario, Canada M3J 1P3
* mhirsch@yorku.ca

## Abstract

Despite the recent resurgence in research on human curiosity and its influence on knowledge acquisition, there is little consensus on how to operationalize curiosity. Understanding lay beliefs on what curiosity is—and how it is experienced—is therefore an important avenue of study. Furthermore, given age-related shifts in emotional processing and goal-seeking, definitions and experiences of curiosity may fluctuate over the lifespan. Here, we used qualitative techniques to characterize perceptions of curiosity in younger and older adults. We found that while both younger and older adults described curiosity as a form of joyous exploration, younger participants more commonly highlighted the link between curiosity and feelings of deprivation, thrill-seeking, and centrality to being human. Older adults, however, more frequently referenced the social aspects of curiosity, and interestingly, approximately 50% of all participants' responses were categorized as miscellaneous, suggesting that current multidimensional frameworks of curiosity may not completely align with lay conceptualizations. Older adults were also more likely than younger adults to describe curiosity as a positive trait, indicative of one's sincerity. In contrast, younger adults more frequently described curiosity as a trait linked to novelty-seeking and personal growth, but that it should be expressed in moderation, likely because of its potentially harmful consequences (e.g., invading others' privacy). Overall, we show that conceptualizations and experiences of curiosity vary between younger and older adults—an important empirical contribution given the trait's ubiquity across the lifespan and its role in promoting opportunities for lifelong learning.

## Introduction

The past fifteen years have seen a resurgence in research on human curiosity and its influence on knowledge acquisition [1]. However, there is little consensus in the literature on how curiosity should be operationally defined. Early theorists conceived of

**Data availability statement:** All relevant data are available within the paper, its Supporting Information files, and on Open Science Framework (osf.io/vqxfr). A package containing examples, templates, and scripts for conducting qualitative content analysis is available on Open Science Framework (osf.io/uav2q).

**Funding:** Natural Sciences and Engineering Research Council of Canada Award Number RGPIN-2019-06296 to A.-A.C. (https://nserc-crsng.canada.ca/en). The funders had no role in study design, data collection and analysis, decision to publish, or preparation of the manuscript.

**Competing interests:** The authors have declared that no competing interests exist.

curiosity as the 'itch to know' that we feel when faced with uncertainty. Both Berlyne [2] and Loewenstein [3] suggested that curiosity is motivated by a feeling of deprivation, which the individual is driven to resolve via information-seeking behaviours (e.g., looking up the answer to a burning trivia question or trying to decipher what someone is saying in a noisy environment). Litman [4,5] suggested that curiosity can also be motivated by enjoyment and pleasure and distinguished between deprivation curiosity, which is driven by frustration at not knowing, and interest curiosity, which is driven by the desire to explore and expand our knowledge.

More recently, curiosity has been described through a multidimensional lens. Kashdan et al. [6] point out that curiosity varies as a function of the specific life contexts in which it is applied (e.g., curiosity for social versus educational information) and in terms of its relationship to other personality traits (e.g., novelty seeking, openness to experience, sensation seeking, need for cognition). They developed the Five-Dimensional Curiosity Scale (5DC; [6]) in an attempt to synthesize these various conceptualizations of curiosity, and outlined five dimensions: joyous exploration (i.e., desire to seek out new information as a means of experiencing joy), deprivation sensitivity (i.e., desire to seek out information as a means of reducing information gaps), stress tolerance (i.e., willingness to embrace uncertainty while acquiring information), social curiosity (i.e., desire to seek out information about others), and thrill-seeking (i.e., willingness to take risks to acquire information). Litman et al. [7] described yet another curiosity dimension—intrapersonal curiosity—which represents one's curiosity for information about oneself.

Thus far, empirical notions of curiosity remain almost entirely theory-driven. Existing scales were constructed by researchers based on their own conceptual assumptions, and little is known about how people themselves define and evaluate it. Lay conceptions are psychologically consequential: people's beliefs about what curiosity is—and whether it is viewed as a positive ("inquisitive," "open-minded") or a negative trait ("prying," "risky")—shape whether they express curiosity and in what domains. Qualitative approaches are therefore essential for identifying aspects of curiosity that may be absent from research-generated constructs, including developmentally significant differences. Indeed, one reason why curiosity may be difficult to articulate is that its expression changes over the lifespan. Children are seen as invariably curious, and acquiring knowledge is necessary for their cognitive and personal development. As we age, however, curiosity may become more person and context-dependent. Studies of trait curiosity suggest that curiosity declines with age ([8–11]; cf. [12]), though recent work employing the 5DC found that not all aspects of curiosity decline: Older adults showed higher joyous exploration and stress tolerance, and comparable deprivation sensitivity relative to younger adults [13]. By contrast, studies of state curiosity, which represents one's momentary experiences of curiosity in a given context [8], show stability [11]—or even an increase [8,10,11,14]—in curiosity with age. The findings of a recent study [8,14] illustrate this paradox: The authors found that both younger and older adults' curiosity was similarly piqued by difficult trivia questions, but that older adults reported themselves to be less generally curious on a self-report measure of trait curiosity relative to younger adults.

Overall, findings show that, while older adults' curiosity is just as likely to be excited by environmental stimuli as that of younger adults, they are less likely to report curiosity when reflecting on their own traits. In a recent study [13], we suggested that both younger and older adults may harbour the perception that curiosity declines with age. If internalized, this perception could lead older adults to self-report lower levels of curiosity, despite typically expressing it. Age differences in the experience and valuation of curiosity may also emerge because curiosity asks people to approach uncertainty, and uncertainty becomes differently consequential across the lifespan. Socioemotional Selectivity Theory (SST) posits that, with shorter time horizons, one's focus shifts away from novelty and information acquisition towards satisfying emotional goals [15]. Through this lens, uncertainty may become more threatening both cognitively and socially as we get older. If conceptualizations of curiosity change over the lifespan, these shifts may have consequential implications for *how* curiosity is measured, raising the question of whether current measures accurately capture what it means to be curious for younger and older adults alike.

The aim of the present study was to explore younger and older adults' perceptions of what curiosity is and whether it is viewed as a positive or a negative trait. Understanding these beliefs is critical, as mismatches between researcher-defined and lay-defined curiosity may obscure how and when people actually choose to engage with uncertainty. Few studies have explored lay conceptions of curiosity (see [16–18]), and none have compared how adults across the lifespan conceptualize and evaluate it. We addressed these gaps using qualitative methods, whereby participants responded to open-ended questions that gauged their definitions of and attitudes towards (i.e., valence) curiosity. Given the exploratory nature of the present work, we did not hold predictions concerning the specific ways in which conceptualizations would differ by age. Although, given mixed findings on what happens to curiosity with age (e.g., [8]) as well as evidence for age-related shifts in emotional processing and goal pursuit [19], we expected to find age differences in conceptualizations, more generally, both in terms of the definitions and the valence descriptions provided.

## Materials and methods

### Participants

A total sample size of 128 was determined a priori using the *pwr* package in R [20,21], requiring a power of at least 80% to detect a medium-sized effect (Cohen's $d = 0.50$), assuming $\alpha = .05$. This corresponded to 64 participants per group (younger and older adults) for independent-samples *t*-tests, consistent with the analytical approach taken in Hirsch et al. [13]. To accommodate exclusions, between March and April of 2025, we recruited 95 and 100 English-speaking younger (aged 18−35 years) and older (aged 60−99 years) adults through the York University Undergraduate Research Participant Pool and the Glendon Research Participant Pool, respectively (i.e., a convenience sampling approach was used). The Glendon Research Participant Pool consists of community dwelling older adults that consented to be contacted about studies conducted at Glendon College. Participants were excluded if they: showed response biases on questionnaires (i.e., little to no variability across questionnaire items; identified via manual inspection of response patterns by the first author); spent less than 14 minutes completing the study (i.e., haphazardly submitted responses); did not pass all attention checks; reported using Artificial Intelligence (AI) to generate responses; and/or indicated that their responses did not accurately reflect their thoughts and/or opinions. After exclusions, the final sample consisted of 75 younger and 80 older adult participants (see Table 1 for sample demographics). Younger adult participants were compensated with course credit while older adults were compensated monetarily ($10.00 CAD). The present study received approval from the York University Office of Research Ethics (certificate number 2024−294).

### Materials

Study materials included the following. Two open-ended questions were created by members of the research team. The first question, hereafter referred to as the *Definition Question*, was "*In your opinion, what does curiosity mean (i.e., What*

**Table 1. Sample characteristics and age group differences.**

| | Younger adults M (SD) | Older adults M (SD) | Difference t |
|---|---|---|---|
| N | 75 | 80 | – |
| N (female) | 58[a] | 63 | – |
| Age | 19.4 (1.56) | 72.2 (6.33) | – |
| Age range | 18-26 | 62-94 | – |
| Race (White) | 16 | 70 | – |
| Education | 12.81 (2.59) | 16.35 (2.48) | 8.67** |
| English | 63 | 74 | – |
| SES ($k) | – | – | – |
| <60 | 28 | 26 | – |
| 60-120 | 32 | 35 | – |
| >120 | 15 | 19 | – |
| SILS | 29.76 (4.72) | 36.85 (2.28) | 11.79** |
| HADS | – | – | – |
| Anxiety | 12.12 (3.68)[b] | 19.05 (4.11) | -5.02* |
| Depression | 11.50 (1.60)[b] | 4.53 (2.62) | 10.93** |
| PANAS | – | – | – |
| Positive | 29.12 (8.01) | 33.33 (6.67) | -3.54* |
| Negative | 21.19 (8.91) | 11.96 (4.10) | 8.19** |

t values correspond to two-tailed t test results for age group differences. Age and education are measured in years and the Shipley Institute of Living Scale (SILS), Hospital Anxiety and Depression Scale (HADS), and Positive and Negative Affect Schedule (PANAS) are measured as scores. Socioeconomic status (SES) is measured in Canadian dollars. "English" represents the number of participants that learned English before the age of five.

* $p < .001$. ** $p < .0001$.

[a] Transgender ($n = 1$).

[b] $N = 8$.

is your definition of curiosity? What does it mean to be curious?)?" The second question, hereafter the *Valence Question*, was "*Do you think of curiosity as more of a positive or negative trait? Please elaborate on your response.*" The Shipley Institute of Living Scale (SILS; [22]), Hospital Anxiety and Depression Scale (HADS; [23]), and Positive and Negative Affect Schedule (PANAS; [24]) were included as measures of vocabulary, trait, and state mood, respectively. A demographic questionnaire, a question asking whether participants used AI to aid in their responses (yes/no), and a question asking whether participants' responses accurately reflected their thoughts and opinions (yes/no) were also included. Original and adapted versions (see osf.io/vqxfr) of the Five-Dimensional Curiosity Scale (5DC; [6]) and Intrapersonal Curiosity Scale (InC; [7]) as well as an additional open-ended question (i.e., asking participants about what happens to curiosity with age) were also administered, although are not relevant to the aims of the present study (for details, see [13]).

## Procedure

The present study was delivered online using Qualtrics [25]. After providing written informed consent, participants completed a demographic questionnaire, the SILS, and original versions of the 5DC and InC. Participants then provided typed responses to the open-ended questions. To encourage substantive responses, each open-ended question required responses that were at least 250 characters in length (~ 40 words). Finally, participants completed adapted versions of the 5DC and InC, the HADS, the PANAS, and questions regarding AI use and whether responses accurately reflected their thoughts and opinions. Materials were presented in the same order for all participants.

## Content analysis

Across both samples (i.e., younger and older adults), hybrid qualitative content analysis was used (see [26]). A hybrid content analysis approach allowed us to utilize both deductive (top-down method, using pre-established coding schemes to code responses) and inductive (bottom-up, data-driven approach used to develop codes) methods to conceptualize open-ended response data [27,28]. For the *Definition Question*, deductive analysis was applied using the 5DC and InC coding scheme [6,7]. Accordingly, responses were coded into one or more of the following categories: *joyous exploration*, *deprivation sensitivity*, *stress tolerance*, *social curiosity*, *thrill seeking*, and *interpersonal curiosity* (for descriptions of each dimension, which were taken from 5DC and InC questionnaire items, see Table 2). Any responses that did not fall into any of these categories were categorized as *miscellaneous*. Responses were coded such that a given participant's response could be coded as more than one of the above categories.

Inductive analysis was used for the *Valence Question*, which was conducted in two stages. First, each participant's initial response was coded as reflecting curiosity as a *positive* or a *negative trait*. In addition, responses could be coded as *in moderation* when participants specifies that curiosity should be expressed to a limited degree. Thus, initial responses were assigned either a single code (*positive* or *negative trait*) or, in some instances, two codes (*positive trait* and *in moderation* or *negative trait* and *in moderation*). Next, the rationale underlying each response was coded separately. Multiple rationale codes could be assigned to a single response when it reflected more than one coding category, regardless of response length. Potential rationales for initial responses included the following categories: *motivated learning*, *critical process*, *novelty-driven*, *advance knowledge*, *personal growth*, *harmful*, *sincerity*, *individual differences*, *centrality* and *miscellaneous* (i.e., for rationale that did not fall into any of these categories). Initially, such categories were inductively derived as follows. Two primary coders independently reviewed 50% of the responses and began grouping responses into coding categories based on observed patterns, iteratively developing and refining category labels. Following this initial review, the authors met to discuss and reconcile category definitions, revising category labels and descriptions until consensus was reached.

## Inter-rater reliability

After consensus was achieved, the same two primary coders each coded 50% of the full dataset. To assess reliability, a third coder coded a random 25% of all participant responses, ensuring validity of the coding schemes across questions (see [29]). Percentage agreement and Cohen's kappa indices were calculated to assess inter-rater reliability (see OSF; [30]). Coefficients of .80 or greater reflect acceptable agreement [29], and thus if reliability scores fell below this threshold, coders met to discuss and resolve disagreements, consulting the senior author (A.-A.C.) if agreements were not met. Inter-rater reliability was very high across coders. For the younger adult sample, the *Definition Question* (percentage agreement ≥ 95%; κ ≥ 0.83), and the *Valence Question* (percentage agreement ≥ 95%; κ ≥ 0.88) reached high reliability scores. Likewise, inter-rater reliability for the older adult sample reached high reliability for both the *Definition Question* (percentage agreement ≥ 95%; κ ≥ 0.89), and the *Valence Question* (percentage agreement ≥ 95%; κ ≥ 0.83).

## Results

All statistical analyses were conducted using R (v4.4.1; [21]). To understand what people think curiosity is, and how these definitions and experiences of curiosity vary by age, we employed two open-ended questions. For the *Definition Question,* we deductively coded participant responses into pre-established categories as per the separable dimensions of curiosity defined by the 5DC and InC ([6,7]; note that results are presented in the order in which their respective curiosity dimensions are listed in the 5DC and InC). Frequencies and example participant responses are reported for both samples (see Table 2; Fig 1). For the *Valence Question*, we report on inductively coded categories, whereby participant responses were evaluated for thematic qualitative content, and category codes were

**Table 2. Categories, Descriptions, Examples and Code Frequencies for the Definition Question.**

| Categories | Description | Younger Adults | | Older Adults | |
|---|---|---|---|---|---|
| | | Example | Freq. % | Example | Freq. % |
| 1. Joyous exploration | One views challenging situations as an opportunity to grow and learn. One is always looking for experiences that challenge how they think about themself and the world. One seeks out situations where it is likely that they will have to think in depth about something. One enjoys learning about subjects that are unfamiliar to them. One finds it fascinating to learn new information. | "Curiosity to me means the exploration or excitement in understanding something you do not know due to your own wonder about something as if you are asked to go somewhere new a person would be excited or want to know where or what that place is that is what is called being curious" "In my opinion, curiosity is the natural drive to investigate, discover, and comprehend the world we live in. It involves posing queries, looking for solutions, and remaining receptive to novel experiences even when those experiences or solutions contradict our preconceived notions. To be curious means to delve deeper to find more details, connections, and nuances rather than settling for a superficial understanding" | 90.67% | "Wanting to find answers to questions whether they be very basic things like who starred in a specific movie to move deep questions such as how AI works." "Being interested in learning about something new or never done before, the whys and hows and whats always wanting to know about new people and ways and ideas and customs and history always remaining curious about life and ideas, never stop wanting to know [mire] about something I didn't know before" "It means being interested in almost everything and wanting to understand the world. It means wanting to have new experiences..." | 96.25% |
| 2. Deprivation sensitivity | Thinking about solutions to difficult conceptual problems can keep one awake at night. One can spend hours on a single problem because they just can't rest without knowing the answer. One feels frustrated if they can't figure out the solution to a problem, so they work even harder to solve it. One works relentlessly at problems that they feel must be solved. It frustrates one not having all the information they need. One might mention curiosity in reference to a need, drive, compulsion, strong/ robust desire, pull, longing, and/or urge. One may also allude to curiosity and the search for knowledge as being a relentless and/or persistent act/process. | "It [curiosity] is almost a hate for unsolved puzzles or problems. Being curious means something drives you to answer a question or complete something [incompleted], which can be exciting and frustrating at the same time." "Curiosity is when there's something in the dark and you want to know what's there, or when there's a puzzle that interests you." "After realizing you may not know something others know it creates an intense feeling of curiosity to be apart of this knowingness of information." | 45.33% | "...something will arise that will make me explore that subject even further. Almost like a scientific experiment where I perform an experiment and I'm satisfied with it but then along comes another situation that throws it into a different light and it has me asking why did it happen that way how did it happen and what does this mean." "I think it's a strong desire to know or learn something new." "The drive to learn..." "Curiosity. Is the reaction I have when I don't understand what is happening or going on." | 10.00% |
| 3. Stress tolerance | The smallest doubt can stop one from seeking out new experiences. One cannot handle the stress that comes from entering uncertain situations. One finds it hard to explore new places when they lack confidence in their abilities. One cannot function well if they are unsure whether a new experience is safe. It is difficult to concentrate when there is a possibility that one will be taken by surprise. | "...see[ing] challenges and change as [opportunties] to grow and explore rather than avoiding them." | 1.33% | "Looking for out of the ordinary and unexpected." "Even being curious about world events to be better informed is a good thing although it can be stressful. Unhealthy curiosity can lead to anxiety if you find out things that stress you." "looking under the cover" | 6.25% |

*(Continued)*

|  |  | Younger Adults |  | Older Adults |  |
|---|---|---|---|---|---|
| 4. Social curiosity | One likes to learn about the habits of others.<br>One likes finding out why people behave the way they do.<br>When other people are having a conversation, one likes to find out what it's about.<br>When around other people, one likes listening to their conversations.<br>When people quarrel, one likes to know what's going on. | "For example, if I am curious about how a certain person behaves, I will connect and interact with them- therefore satisfying my curiosity."<br>"People that are around me tend to call me "nosey" because I love to know what people are doing, talking about, thinking, etc."<br>"Some people are more intrigued by human emotions, like why people act the way they do." | 24.00% | "To me it means that I want to find out more about something or someone....I would research if it is a person I do not know – e.g., an expert in a particular field, someone who is famous. If I am curious about someone I meet or know but would like to know better, I would ask them questions in person to get to know them better."<br>"...being open to learning new things about people..."<br>"...some people think that being curious is being 'nosey' that is wanting to know other peoples business."<br>" It means you're interested in finding out more about a person, place or thing. You want to go deeper. for example if you meet someone you're curious about, you want to know more about them." | 43.75% |
| 5. Thrill seeking | The anxiety of doing something new makes one feel excited and alive.<br>Risk-taking is exciting to one.<br>When one has free time, they want to do things that are a little scary.<br>Creating an adventure as one goes is much more appealing than a planned adventure.<br>One prefers friends who are excitingly unpredictable. | "Curiosity is the will to explore new experiences despite the risks that come associated with them."<br>"Being curious to me, is a feeling of anxious excitement....An unsettling thrill which keeps me constantly thinking."<br>"I think curiosity means the thrill to know something new and explore new things in life..." | 10.67% | "Curious people are risk takers." | 1.25% |
| 6. Intra-personal curiosity | One wonders about their purpose in life.<br>One imagines what my life might have been like had they taken different paths.<br>One tries to make sense of how they feel.<br>One finds themself thinking about the reason for their existence.<br>One questions whether they really know who they are. | "...learn and discover more about ourselves...and [curiosity] helps us grow and develop into the people we are meant to become."<br>"For instance, [curiosity is] wondering how the universe works, what happens when we die, or why we were put on this Earth in the first place." | 8.00% | "curiousity is about learning how things work, why people (myself included) make the decisions they do."<br>"Curiosity is what motivates, fuels and entices individuals and groups to continue living." | 3.75% |
| 7. Miscellaneous | Other responses that do not fall into any of the above categories. |  | 50.67% |  | 50.00% |
| *Sub-codes* |  |  |  |  |  |
| Future thinking | One discusses curiosity in relation to future-oriented thinking. This may include mentions of possibilities or potential outcomes. | "[Curiosity is] exploring outcomes or possibilities that could happen. For example when you go to the sample store everyday but you always wondered what was in the next floor so you decided to go up to the floor and explore." | *5.33%* | "Will I be [*positivlely*] or negatively impacted or will the knowledge learned be of use to me."<br>"If you are curious you can not be concerned about outcomes and must be accepting of the out come and remain open to the out come..." | *5.00%* |
| Centrality | One discusses curiosity in relation to its centrality/innateness/importance to life and/or being human. | "I think curiosity is something all humans have..." | *10.67%* | "I believe that you never stop learning in life." | *2.50%* |

*(Continued)*

**Table 2.** (Continued)

| | | Younger Adults | | Older Adults | |
|---|---|---|---|---|---|
| Individual differences | One discusses curiosity in relation to individual differences in its expression/ manifestation. | "…it's [curiosity] just varies how much you use it or how much it is apparent in you." | *24.00%* | "In particular as a nuclear physicist I continue to wonder a out dark energy. I believe that there is something wrong with our basic understanding when we have to include dark energy and or dark matter to explain how the universe works. " | *35.00%* |
| Innovation | One discusses curiosity in relation to advancing individual/society-level knowledge, expressing creativity, and/or adapting to change. | "It [curiosity] pushes individuals to break free from their comfort zones, engage with fresh concepts, and keep growing. Curiosity drives everything from scientific discoveries to artistic creativity and helps us grasp different viewpoints. It's the key element that propels knowledge and progress ahead." | *12.00%* | "To be curious is to enjoy wonder and learning new things both from a practical (useable) perspective" | *5.00%* |
| Other | Responses that do not fall into any of the miscellaneous sub-codes. | " However, curiosity can also be a bad thing sometimes, like when your curiosity leads you into places you shouldn't be or makes you do things that you shouldn't be doing. For example, a person might be curious about how drugs would feel and try them, this can lead to an addiction and ruin their lives. This is why I believe that curiosity can go both ways and it is good as long as you are curious about the right things." | *6.67%* | "…curiosity is being curious. how to be wonder about something new….To me curiosity is about behavior and emotion." | *5.00%* |

*Note.* Code frequencies are presented as percentages. Responses were coded such that more than one code category could apply to a given response.

then developed to represent the data collected. The *Valence Question* (see Table 3; Fig 2) prompted respondents to first indicate whether curiosity is a positive, a negative trait, or a trait to be expressed in moderation, and were then asked to provide rationale for their initial response. For the *Valence Question*, we present results in the order through which codes were inductively derived from the data. For all questions and category codes where significant differences arose between samples, chi-square ($\chi^2$) analyses are reported (see S1 Table and S2 Table to view complete chi-square test results). See supplementary material (S1 Text; S3 Table; S1 Fig; S4 Table; S2 Fig) for additional descriptive analyses and visualizations.

### Defining dimensions of curiosity

**Joyous Exploration (90.67% Younger Adults, 96.25% Older Adults).** Curiosity was most frequently described as a form of *joyous exploration*. More than 90% of participants from both samples described curiosity as a means to grow, an opportunity to look for new experiences that challenge their worldview, and/or as an enjoyable way to learn about new subjects or material. For instance, one younger adult participant stated "Curiosity is questioning, wondering, and seeing the world as something worth exploring. It means seeking knowledge and asking deeper questions, and even if they aren't deep, it's simply questioning everything itself.". Another participant responded:

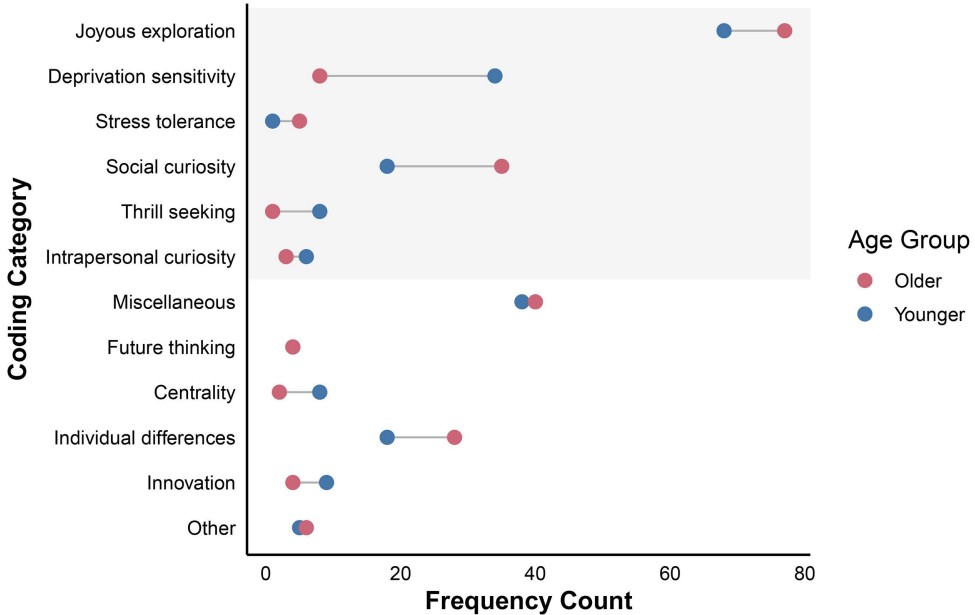

**Fig 1. Frequency Counts for Individual Coding Categories: Definition Question.** *Note.* Frequency counts for each coding category are shown, illustrating how curiosity is defined by younger versus older adults. The shaded region corresponds to coding categories that were deductively derived (using the 5DC and InC), and the non-shaded region corresponds to coding categories that were inductively derived (miscellaneous and its sub-category codes).

> Curiosity...drives us to explore, learn, and understand the world around us. It's the innate desire to seek new knowledge, ask questions, and solve problems. At its core, curiosity is about an openness to the unknown and a willingness to challenge our current understanding. It compels us to ask 'why,' 'how,' and 'what if,' pushing us beyond the boundaries of our comfort zone to discover new perspectives. Curiosity can manifest in many forms: intellectual curiosity, which fuels our pursuit of knowledge and new ideas;...and even physical curiosity, such as exploring unfamiliar environments. ... It enriches our lives by connecting us to new experiences, broadening our horizons, and encouraging lifelong learning. Without curiosity, we would be stuck in the same patterns, missing out on the wonders and opportunities that life has to offer. (Younger Adults).

Older adults described curiosity similarly. For example: "Approaching the world with wonder is curiosity. To ask why, what,…when and how as part of how you approach your life observations and experiences. The most useful question is why. Being curious makes one's life lots of fun!". Additionally, an older adult participant indicated:

> Curiosity is the exploration of your environment to seek new information. It includes critical thinking and problem solving skills to find answers to inquiries. Curiosity is a sense of seeking and growing intellectually. It means wanting to challenge yourself. …a fundamental part of curiosity is learning (Older Adults).

**Deprivation Sensitivity (45.33% Younger Adults, 10.00% Older Adults).** While only 10% of older adults described curiosity as *deprivation sensitivity* (i.e., a need or compulsion that drives the relentless search for knowledge to satisfy curiosity), nearly half of younger adults described curiosity as such. Younger adults reported this code more often than their older adult counterparts, $\chi^2(1) = 24.46$, $p < .001$, $V = .40$, indicating that they may associate curiosity with an experience of frustration until they reach a solution or 'fill in' the purported gap. For instance, a younger adult participant

**Table 3. Categories, Description, Example, and Code Frequencies for the Valence Question.**

| Categories | Description | Younger Adults | | Older Adults | |
|---|---|---|---|---|---|
| | | Examples | Freq. % | Examples | Freq. % |
| ***Initial Response*** | | | | | |
| 1. Positive trait | Individual describes curiosity as a positive trait. | "I believe that curiosity is a positive trait…" "I think of curiosity as a primarily positive trait." | 73.33% | "I think it is a hugely positive trait and enriches one's life." "I think that curiosity is always positive…" "I see it as a positive because I think that alternative is very limiting." | 91.25% |
| 2. Negative trait | Individual describes curiosity as a negative trait. | "…but it [curiosity] can be negative trait…" | 1.33% | "This is very negative…" | 1.25% |
| 3. In moderation | Individual describes that curiosity could be considered both a positive and/or a negative trait, i.e., depending on how curiosity manifests. | "I think it [curiosity] is both equally positive and negative depending on the context." "I think this [curiosity] depends on the person and their personality." | 44.00% | "But there is a limit and risk to always being curious…" "I think it can be both." "Mostly positive. Some negatives for sure…" | 17.50% |
| ***Rationale*** | | | | | |
| 1. Motivated learning | Individual expresses affectively-related thoughts on curiosity relating to a motivation or desire to learn in a general capacity. This is not discussed in the context of specific information/experiences, but instead, there is an open-mindedness to continue to learn. Curiosity provides opportunities to learn interesting things and/or prevent experiencing boredom. | "it drives learning" "…learning more about the world, [eductation], politics, whatever it is will always be more beneficial…" "…it [curiosity] makes individuals want to learn more and get a better understanding of certain situations…" | 50.67% | "It opens up your mind to new possibilities and ways of thinking." "By being curious you expand your horizons and live life more fully. It's a big world and if you're not curious you miss out on what the world has to offer. You only have one life – and curiosity helps you make the most of that life" "…I think an enquiring mind is vital to maintaining an interest in living. Around every corner you might meet an interesting person…or get that light bulb flash over your head when you finally understand a concept that has been alluding you." | 46.25% |
| 2. Critical process | Individual expresses age-related changes/stability in curiosity as a means of questioning things, and attempting to understanding the "whys" and "hows" or the way things "work." Individual may also express age-related changes/stability in curiosity in relation to future-oriented thoughts/thinking. | "…always questioning things…" "It [curiosity] encourages individuals to ask questions…" "…ask questions, and seek deeper understanding…" | 25.33% | "It can provide information, solutions and lead to additional questions." "For simple things, one can ignore/be satisfied with less information and still consider your curiosity satisfied. More complex situations are better understood or less stressful when curiosity provides answers to previous unknowns." "One need to learn to assess, evaluate and direct how one approaches that which is curious." "It teaches us to question and not just accept the status quo." | 17.50% |
| 3. Novelty-driven | Individual expresses affectively-related thoughts on curiosity relating to a continued desire to explore, experience, learn, and/or participate in new things. | "…seek new experiences." "It's [curiosity] apart of human instincts to be curious of things and the unknown certainty." "It [curiosity] helps people explore new ideas… | 37.33% | "…because we now have time to learn new things…" "If you're curious you're more [intersted] in learning new things, trying new things, doing new things, feeling new feelings." "I enjoy learning new ways of doing things or tricks on how to do things more efficiently." "And finding out new things (large or small) is pleasing." | 17.50% |

*(Continued)*

| Categories | Description | Younger Adults | | Older Adults | |
|---|---|---|---|---|---|
| | | Examples | Freq. % | Examples | Freq. % |
| 4. Advance knowledge | Individual expresses affectively-related thoughts on curiosity in the context of innovation, i.e., a way to advance individual/society-level knowledge or tools. An individual may also describe curiosity as a means by which we adapt to new changes and/or challenges or express creativity. | "It [curiosity] drives...discovery, allowing people to challenge their assumptions..." "We wouldn't be so far in society. Finding cures for illnesses and dreams. From biblical men walking in sandals to riding in planes. Curiosity is really important as it causes growth." | 37.33% | "[Curiosity] forms the basis for learning, applying, adjusting to same and different scenarios and relearning." "Thinking of inventions and innovations that have been made over the centuries, I think the underlying trigger is the curiosity of the creator." "Curiosity spurs people to conduct research and publish their findings. Shared information adds to collective awareness." "...it opens up so many pathways to expanded knowledge, awareness and communication skills." | 31.25% |
| 5. Personal growth | Individual expresses affectively-related thoughts on curiosity in the context of personal growth, where curiosity can be used to develop new skills or behaviours, and/or to gain a deeper understanding about oneself (past and/or present selves). | "...it [curiosity] symbolizes and strengthens intelligence." "It [curiosity] also causes development into self realizations. We find out who we are through curiosity." "...it [curiosity] drives...personal growth." | 40.00% | "Curiosity is the fuel of growth." "...but overall the learnings are positive and are a key part of personal growth and often life changing." "Perhaps as one ages one realizes that you have a finite time..." "It allows you to grow mentally..." | 16.25% |
| 6. Harmful | Individual expresses affectively-related thoughts on curiosity in the context of causing harm to oneself or others depending on how curiosity manifests. | "...unchecked curiosity can lead to risks, like invading privacy or pursuing knowledge without ethical consideration." "Being interested in the wrong topics is what would make this trait negative. Curiosity in others lives can lead to further bad habits..." | 49.33% | "...if you become obsessive about the subject." "It can be negative if you show disapproval for what a person believes or does...And if someone has an opinion of something, but no knowledge of the subject, it is also negative." "Without curiosity one would become isolated from everything and everyone. You cannot interact with the world around you without first being curious about something in it." | 21.25% |
| 7. Sincerity | Individual expresses affectively-related thoughts on curiosity in the context of curiosity acting as an indicator of an individual's 'sincerity' or character as they interact/engage with others and the world around them. | "...it [curiosity] means you're paying attention and are engaging in the life around you." "Curiosity helps you understand the other person better. It helps you stay engaged with the world..." | 10.67% | "I believe curiosity opens communication with others with more honesty..." "...[curiosity] demonstrates an active engagement [and] can lead to understanding and compassion." "I can't imagine not being curious, since it shows a lack of interest in almost everything." | 22.50% |

*(Continued)*

 

**Table 3.** (Continued)

| Categories | Description | Younger Adults Examples | Freq. % | Older Adults Examples | Freq. % |
|---|---|---|---|---|---|
| 8. Individual differences | Individual expresses affectively-related thoughts on curiosity in the context of individual differences. This means that whether curiosity is considered a positive or negative trait depends on a given individual, how it manifests in them, and/or how an individual plays an active role (i.e., agency) in expressing their curiosity. | "So a person chooses whether they want their curiosity to be either positive or negative." "...nothing wrong with having a sense of curiosity. All humans behave, look, and speak differently and that is the reason for curiosity." "...this [curiosity] depends on the person..." "I think it [curiosity] depends how much curiosity you have and also how you manage it." | 20.00% | "It is hard for me to imagine what my life would be like if I were not curious about the things around me...I worked for 20 years in high tech as a technical writer, I was curious and enjoyed learning about the "gizmos" I was writing about in user guides and quick start guide. For a short time I worked for a company that made income tax software and documented it in user guide. I was not at all interested or curious about income tax policies. I found that particular job incredible boring and did not last. Without curiosity about the subject I was writing about, the job was dull." "For me curiosity has allowed me to explore an artistic side which in turn has led me to explore nature with more curiosity and a different way of seeing the world around me. Curiosity has led to me being a volunteer ... where I have learned about the importance of native turtles in our area and actively work to protect nesting sites and educate the public." "If people are able remain in an environment that support seeking answers then I believe curiosity is positive. Those who are able to stay connected to others has an environment that can support curiosity" "If you are a curious person you would always be interested in your surroundings or learning..." | 22.50% |
| 9. Centrality | Individual expresses affectively-related thoughts on curiosity in the context of its centrality/innateness/ importance to life and/or being human. | "I feel our brains are wired to be curious about things and our lives..." "We simply cannot help it." "...it [curiosity] helps you...find meaning in your life." "Learning is essentially for us as a species..." | 12.00% | "If you stop learning it's almost like not living." "...I believe curiosity is a fundamental human trait. I believe that every human being wants to learn to stay alive." "Curiosity engages us in the world around us. It should be part of everyday life." | 5.00% |
| 10. Miscellaneous | Other responses that do not fall into any of the above categories. | "Like are you only focused on other people or do you actually have a life." "It's also very good to maintain good cognitive health, which is definitely a plus as we grow older." "You can be curious about what your mom might cook for dinner, but you won't be disappointed knowing that you would like whatever she would make." | 10.67% | "No such thing as too much information." "Curiosity keeps you mentally active, which is always a good thing." "You would almost be a vegetable if you were not curious." "[Curiosity] can help keep our brains sharp as we age." | 21.25% |

*Note.* Code frequencies are presented as percentages. Initial responses were coded as either *positive trait* or *negative trait*. Initial responses could additionally be coded as *in moderation* (i.e., in addition to being coded as either *positive trait* or *negative trait*). Rationale for initial responses were coded such that more than one code category could apply to a given response.

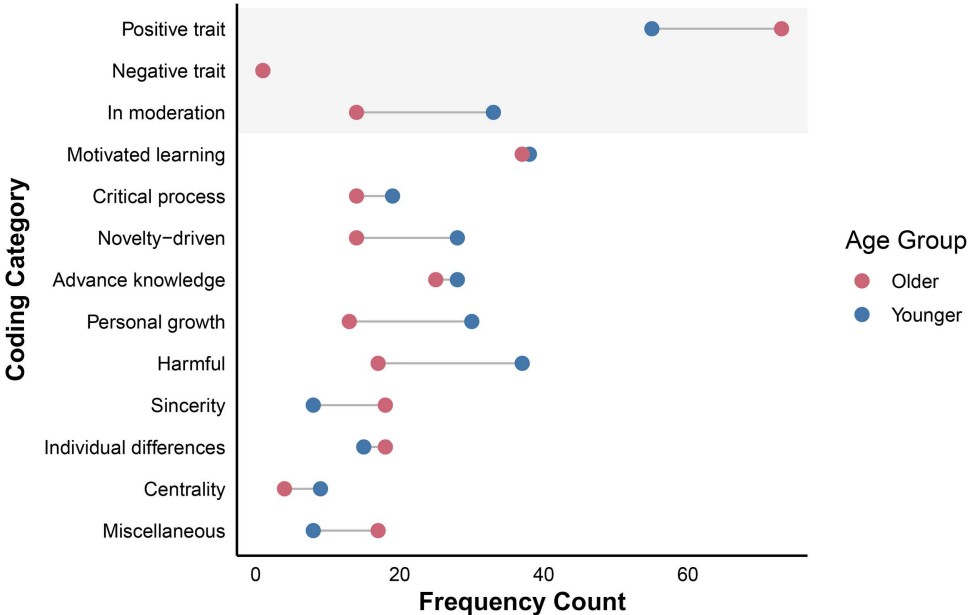

**Fig 2. Frequency Counts for Individual Coding Categories: Valence Question.** *Note.* Frequency counts for each coding category are shown, illustrating how curiosity is experienced (i.e., in terms of its valence) by younger versus older adults. The shaded region corresponds to the coding categories of initial responses, and the non-shaded region corresponds to the coding categories of rationales for initial responses.

described: "Curiosity means a drive to learn about things that a person questions and wants that knowledge to quench their curiosity. Curiosity [is the] pull to learn things…". Curiosity is "a sense of feeling on edge, unsatisfied until I find the answers Im looking for....I feel restless and antsy, but in a positive way. Its a strong desire that needs to be fulfilled so I can move on." (Younger Adults). Although less frequently, some older adults described curiosity in similar terms: "Curiosity is an insatiable desire to dive deeper, to investigate, to learn more, and to know more about a person, place, event, or thing." Put differently,

> Curiosity is a need to know something you don't understand but find intriguing. It doesn't have to be anything specific but happens to trigger a sense of needing to understand it better than you already do (Older Adults).

**Stress Tolerance (1.33% Younger Adults, 6.25% Older Adults).** Only about 1% of younger adults and about 6% of older adults viewed curiosity as a form of *stress tolerance*. That is, few participants discussed curiosity in reference to their abilities to tolerate stress resulting from uncertainty. When they described curiosity in these terms, participants characterized curiosity as "...see[ing] challenges and change as opportun[i]ties to grow and explore rather than avoiding them." (Younger Adults). One participant described curiosity as "...looking behind a curtain or other obstacle blocking one's view" (Older Adults). Another older adult indicated that:

> If you are stressed out and focusing your mind on the thing that is causing you stress as you walk down the street, it is possible to miss all kinds of interesting stimulus that would cause your mind to be not curious at all.

**Social Curiosity (24.00% Younger Adults, 43.75% Older Adults).** One of the next most prevalent responses was that of *social curiosity*. A quarter of younger adults and almost half of the older adult participants indicated that they liked

to learn about other people, for example, by listening to others' conversations and knowing what is going on in their lives. Older adults reported this code more often than younger adults, $\chi^2(1) = 6.71$, $p = .009$, $V = .21$. For instance, "Being curious also means learning new things about the people around you and learning about their habits, likes, and dislikes." and "...if a friend or family is incredibly upset about something, I'm curious to find out why. If someone dies I almost always ask how even if it's considered rude, because I am interested in it." (Younger Adults). One participant stated:

> It means that you are interested in what makes people or things tick. For example why does a person respond to another person in a certain way or why does a person act a certain way in situations. In my own situation I am always curious about people, where did they come from, what kind of an upbringing did they have, what kind of a career did they have, what are they most interested in doing for example their hobbies (Older Adults).

**Thrill Seeking (10.67% Younger Adults, 1.25% Older Adults).** *Thrill seeking* refers to feelings of thrill or excitement about doing things that are unknown or could be considered scary or adventurous. Younger adults reported this code ten times more frequently than their older adult counterparts (roughly 11% versus 1%, respectively), $\chi^2(1) = 6.28$, $p = .012$, $V = .20$. Responses indicated that curiosity is about "going out of your comfort zone and just exploring life to the fullest…" (Younger Adults). Additionally, it refers to the enjoyment of new activities despite potential risks, such as being "... interested in taking risks and trying new things. ... [One] enjoys taking risks and is open to new opportunities." (Younger Adults). Simply put, "Curious people are risk takers." (Older Adults).

**Intrapersonal Curiosity (8.00% Younger Adults, 3.75% Older Adults).** Among younger and older adults, *intrapersonal curiosity* was seldom used to describe what it means to be curious. *Intrapersonal curiosity* reflects instances in which one wonders about their purpose in life or about themselves: "...to learn and discover more about ourselves...and [curiosity] helps us grow and develop into the people we are meant to become." (Younger Adults); "It means being a life long learner in regards to relationships, including with yourself…" (Older Adults). It also refers to how individuals fit into the world around them: "For instance, wondering how the universe works, what happens when we die, or why we were put on this Earth in the first place." (Younger Adults).

**Miscellaneous (50.67% Younger Adults, 50.00% Older Adults).** Interestingly, half of all responses from both samples fell outside of definitions of curiosity provided by the 5DC and InC. Accordingly, they were categorized as *miscellaneous* and were further inductively coded based on their thematic content. Below are the categories that miscellaneous responses were categorized into.

**Future Thinking (5.33% Younger Adults, 5.00% Older Adults).** For both samples, 5% of all responses referred to curiosity as a mechanism of *future thinking* or considering future-oriented thoughts and/or outcomes. Both samples indicated their desire to learn about potential outcomes of situations: curiosity is "...trying to figure out...how things could have been different if a different decision was made." (Younger Adults); "...I'm always curious about how the future will unfold." (Older Adults).

**Centrality (10.67% Younger Adults, 2.50% Older Adults).** Less than 3% of older adults and just over a tenth of younger adults, $\chi^2(1) = 4.28$, $p = .039$, $V = .17$, described curiosity in reference to *centrality* or the inherently human nature of curiosity. Curiosity was described as "...a fundamental human trait...It's the innate desire to seek new knowledge…" (Younger Adults) and as a "...normal part of being alive..." (Older Adults).

**Individual Differences (24.00% Younger Adults, 35.00% Older Adults).** The most frequently used *miscellaneous* sub-category code was that of *individual differences*. About a quarter of younger adults and a third of older adults described curiosity in relation to how it manifests differently for different individuals, for example, "I believe that some people are more curious than others…" (Younger Adults). The sub-code was also exemplified in responses concerning personal interests, for example, "I travel a lot and my head is on a swivel taking in new surroundings" (Older Adults).

**Innovation (12.00% Younger Adults, 5.00% Older Adults).** Over 10% of younger adults, and only 5% of older adults referred to curiosity as a means for *innovation* or a tool to advance knowledge (e.g., "I expect any day that someone will com[e] up with a completely different explanation of how the universe works at both the quantum level and the [*astronomcal*] level."; Older Adults), express creativity, or adapt to change (e.g., "It [curiosity] motivates innovation, creativity, and personal growth....In essence, curiosity is the spark that drives human progress, fostering discovery and helping us adapt to changing circumstances."; Younger Adults).

**Other (6.67% Younger Adults, 7.50% Older Adults).** Lastly, responses that did not fall under any of the developed *miscellaneous* sub-category codes, were categorized as *other* (e.g., "Curiosity is more prevalent in younger years of life, but still remains until the day we die.", Younger Adults; "When life brings you Shit - go find the pony.", Older Adults).

## Describing the Valence of Curiosity

**Positive Trait (73.33% Younger Adults, 91.25% Older Adults).** As discussed above, coding for the *Valence Question* was a two-step process, where initial responses were coded first, followed by coding of rationale for initial responses. The most prevalent initial response to the *Valence Question* was that curiosity could be defined exclusively as a *positive trait*. Three quarters of the younger adult respondents (e.g., "I think of curiosity as more of a positive trait…"), and over 90% of older adults (e.g., "I think it's a positive trait.") described curiosity as a positive or desirable trait, with older adults more likely to define curiosity as positive, $\chi^2(1) = 8.64$, $p = .003$, $V = .24$.

**Negative Trait (1.33% Younger Adults, 1.25% Older Adults).** Only about 1% of both younger (e.g., "...but it [curiosity] can be negative trait…") and older adult (e.g., "This is very negative...") participants provided initial responses indicating curiosity as an exclusively undesirable, or *negative trait* to experience.

**In Moderation (44.00% Younger Adults, 17.50% Older Adults).** Curiosity was also commonly described as desirable, however only when expressed *in moderation*. This code indicates that curiosity is not exclusively positive or negative, but can be either valence depending on how it manifests. Younger adults were more likely to state that curiosity could manifest in either a positive or negative fashion, $\chi^2(1) = 12.87$, $p < .001$, $V = .29$, while older adults responded in this way less than a fifth of the time. For example, a younger adult participant indicated that "Curiosity is generally a positive trait....However, curiosity can have negative aspects". Similarly, an older adult participant noted that curiosity "Can be considered either depending on what individuals are [curious] about.", suggesting that curiosity is not an inherently positive or negative trait but may be context-dependent.

**Motivated Learning (50.67% Younger Adults, 46.25% Older Adults).** After coding initial responses to the *Valence Question*, rationale for such responses were coded. The most prevalent rationale provided was that of *motivated learning*. Half of all younger adult participants, and over 40% of older adults reported curiosity valence (i.e., positive, negative, or in moderation) as a function of one's general open-mindedness to learn new things. This could include the desire to discover new knowledge, "Curiosity invokes learning and discovery, and I think having knowledge can be useful in a variety of ways." (Younger Adults), or finding pleasure in learning, "To be curious and have desire to learn more always seems to increase dop[a]mine levels and I'm always chasing that dop[a]mine high." (Older Adults). Additionally, remaining open-minded to new experiences, where "...Around every corner you might meet an interesting person...or get that light bulb flash over your head when you finally understand a concept that has been alluding you." (Older Adults).

**Critical Process (25.33% Younger Adults, 17.50% Older Adults).** A quarter of younger adults and less than a fifth of older adults described curiosity as a *critical process*, i.e., a means to question the world around them. This includes understanding the "whys" and the "hows" of the world, or understanding the way things "work" (Younger Adults). Put differently, "Without curiosity people would lack the ambition to find answers for questions…" (Younger Adults). It also refers to participants' discussions surrounding future outcomes, such as "...to ask oneself if there is anything that one can

do to prevent a dark event from recurring. On the other hand what one can do to promote a more positive event." (Older Adults).

**Novelty-Driven (37.33% Younger Adults, 17.50% Older Adults).** When considering curiosity's valence, some participants tied their responses to a desire to explore, learn about, and/or participate—not just generally, as in *motivated learning*—but in particular, in novel experiences (i.e., *novelty-driven*). Over a third of younger adults and less than a fifth of older adults described curiosity in relation to engaging with novel material, $\chi^2(1) = 7.71$, $p = .006$, $V = .22$. A younger adult participant expressed how one is "...more likely to try new things, broaden our horizons, and interact meaningfully with the world around us when we are curious.". Similarly, an older adult participant described the inherently novel nature of everyday experiences as "Every experience we have is technically new in some way as we have not lived it before. Curiosity allows us to take advantage to learn about every new experience."

**Advance Knowledge (37.33% Younger Adults, 31.25% Older Adults).** Over a third of younger adults, and slightly less than a third of older adults, defined curiosity's valence as a means by which to *advance knowledge*. Curiosity drives innovation and knowledge gain, promoting one's ability to adapt to new changes or challenges. It is also key in creative problem-solving and discovery ("Without curiosity, how would discoveries ever be made?"; Older Adults). For instance, one participant indicated "People who are curious are better able to push boundaries, pose meaningful queries, and find solutions to issues. It is the cornerstone of advancement in a wide range of fields, including technology, art, science, and even personal growth." (Younger Adults). Across age, participants believed curiosity to be the motivator through which innovation occurs, "It is said 'necessity is the mother of invention.'. If that is true, curiosity is the midwife. Curiosity motivates discovery, learning, and understanding. Without curiosity, innovation and change would [be] less probable" (Older Adults).

**Personal Growth (40.00% Younger Adults, 16.25% Older Adults).** Younger participants were more likely than older adult participants to relate experiences of curiosity to instances of *personal growth*, $\chi^2(1) = 10.89$, $p = .001$, $V = .27$. Specifically, participants indicated that curiosity assists in developing new skills or behaviours, and/or aids in better understanding oneself and growing as a person. One participant remarked how this growth can be in relation to one's career, "...curiosity in workspaces can lead to excelling in positions." (Younger Adults), while others more generally noted curiosity's impact on personal growth, "...helps one to grow and be more resilient." (Older Adults).

**Harmful (49.33% Younger Adults, 21.25% Older Adults).** Roughly half of younger adults viewed curiosity as *harmful*, while only a fifth of older adults held this view. Depending on how curiosity manifests, it can cause harm to oneself, or to others. For example, one participant noted "However when it is…not looked after well enough it can lead to over stepping boundaries or excessive questioning that disrupt our well being." (Younger Adults). Additionally, depending on how curiosity manifests in an individual, it may cause annoyance or provoke negative feelings in others: "...one might annoy another if he/she is always displaying curiousity." (Older Adults). Interestingly, younger adults were more cautious than their older counterparts and believed curiosity could manifest in harmful ways if not expressed mindfully, $\chi^2(1) = 13.45$, $p < .001$, $V = .30$.

**Sincerity (10.67% Younger Adults, 22.50% Older Adults).** Curiosity may act as an indicator of an individual's *sincerity* or character in reference to how they interact or engage with the world around them. While only a tenth of younger adults described curiosity in relation to sincerity, over 20% of older adults expressed how curiosity may reflect one's character, i.e., their sincerity, $\chi^2(1) = 3.88$, $p = .049$, $V = .16$. Through this lens, curiosity was described as the ability to be cognizant and actively engage with the world ("Curiosity is a means of keeping us engaged in and aware of our surroundings."; Older Adults). It was also used as a way of describing one's character, "I feel like curiosity makes us more likeable and friendly as well because when we are genuinely curious, and show interest in other people, their lives, they would probably like us more." (Younger Adults).

**Individual Differences (20.00% Younger Adults, 22.50% Older Adults).** Younger and older adults similarly referred to the *individual differences* that arise when one expresses their curiosity. For instance, one participant noted the idiosyncratic nature of curiosity and why it manifests, "...nothing wrong with having a sense of curiosity. All humans

behave, look, and speak differently and that is the reason for curiosity." (Younger Adults). Participants also expressed personal accounts of how curiosity has led them on a new path in life. For example, one participant noted:

> ...[curiosity] has enriched my life by leading me to engage with new friends, travel to new places and continue my education through reading and researching. At age 40, I decided to write the Mensa exam and audition for a choir. Subsequently, I passed the Mensa test and  ...  performed musicals in community theatre for 13 years (Older Adults).

**Centrality (12.00% Younger Adults, 5.00% Older Adults).**  Only 12% of younger adults, and 5% of older adults described curiosity's valence as a function of *centrality*, i.e., curiosity as an innately human characteristic. A small subset of participants from both samples referred to curiosity as being "...a fundamental human trait…every human being wants to learn to stay alive." (Older Adults). Further, humans' "...brains are wired to be curious about things and our lives…" (Younger Adults) as "Learning is [essential] for us as a species…" (Younger Adults).

**Miscellaneous (10.67% Younger Adults, 21.25% Older Adults).**  Responses were categorized as *miscellaneous* when they did not fall into any of the above rationale categories. Roughly 10% of younger adult responses and over 20% of older adult responses fell into this category. For example: "...[we are] putting more money and focus on less relevant things...rather than spending such amounts on people that actually need it." (Younger Adults); "For one thing I believe the more you use your brain, the better off you are." (Older Adults).

## Discussion

Empirical work on curiosity has yet to reach a consensus on precisely *what* curiosity is. For example, trait curiosity had been described as a desire for specific (i.e., particular) versus diversive (i.e., general) information [2,3], a feeling of knowledge deprivation versus interest [4,5], and a multidimensional trait [6,7]. Concurrently, little is known about what happens to curiosity as we age (e.g., [8]), suggesting that potential differences in how we conceptualize and experience curiosity with age might contribute to this variance. To gain a better understanding of how curiosity is defined and experienced—particularly in the context of aging—the present study deductively coded participant responses to the *Definition Question* using the 5DC [6] and InC [7] curiosity dimensions.

Among both younger and older adults, curiosity was most frequently defined as a form of *joyous exploration* (Table 2), or an interest in learning novel information and/or challenging existing beliefs. However, this finding was not unanimous; other curiosity dimensions were also frequently represented in lay definitions of curiosity. For example, younger adults were more likely to describe curiosity as a form of *deprivation sensitivity*. That is, younger adults more often referenced curiosity as an 'itch' or drive to satisfy knowledge gaps, as evidenced by greater mentions of *deprivation sensitivity*, mapping onto recent work showing higher deprivation sensitivity in younger than in older adults ([31]; cf. [12]). Older participants, by contrast, more frequently viewed the trait through a social lens. They were significantly more likely than younger adults to reference *social curiosity*, running counter to previous findings that older adults report lower social curiosity than their younger counterparts [9,13]. The present findings thus suggest that, despite having relatively lower curiosity for social information than younger adults, older adults still conceptualize curiosity as highly social in nature. Younger adults also frequently associated curiosity with *thrill-seeking* behaviours, which makes sense in light age-related decreases in openness to experience [32], a personality trait tied to novelty seeking.

Interestingly, approximately 50% of the definitions of younger and older adults were coded as *miscellaneous*, indicating that they did not fall within any of the six pre-specified curiosity dimensions. This finding may signal that established multidimensional frameworks do not capture all the key dimensions of lay definitions of curiosity. In fact, people reliably emphasized how variable it was across people: The most frequently reported *miscellaneous* sub-code was *individual differences*, defining curiosity as something that varies from person to person. So, curiosity might be defined by its heterogeneity, both in terms of the construct itself and variability across people. Another sub-code, *centrality*, was more frequently used by

younger adults than by older adults: Younger adults more often considered curiosity central to the human experience—a description also observed in Aslan et al. [16]. For example, when participants were asked to define curiosity, they frequently described it as an innate feeling which motivates and supports learning [16]. Accordingly, because earlier stages of life involve more rapid knowledge accumulation and exploration, younger adults may view curiosity as imperative to their personal development.

## Experiences of Curiosity Change Over the Lifespan

We sought to explore whether the valence of the experience of curiosity also changes over the lifespan. To gauge attitudes towards curiosity—and how they might vary with age—we inductively coded participant responses to the *Valence Question*, asking participants whether they view curiosity as a positive trait, a negative trait, or a trait to be expressed in moderation (Table 3). Although most participants described curiosity as a positive rather than a negative trait (consistent with Aslan et al. [16]), older adults were significantly more likely to do so, suggesting the potential influence of an age-related positivity effect [19]. If older adults orient their attention towards activities that maintain a positive affect, they may be more likely to deploy curiosity for more positive over negative stimuli. Therefore, they may feel that their experience of curiosity is subjectively more positive. Indeed, older adults were less likely than younger adults to associate curiosity with deprivation in this sample (i.e., a curiosity facet linked to negative affect).

Among younger adults, describing curiosity as positive was most commonly linked to its ability to promote *motivated learning*, foster *personal growth*, engage with novel stimuli (i.e., *novelty driven*), and *advance knowledge* (S3 Table). Similarly, older adults used *motivated learning* and *advance knowledge* to rationalize curiosity as a positive trait. They also rationalized using *miscellaneous* (for example, one participant stated, "I think of curiosity as a positive trait in a person's health and well-being lifestyle.") and *sincerity* (i.e., curiosity acts as an indicator of an individual's sincerity as they interact with others and the world around them; S4 Table). For example, an individual that asks questions about others and engages in active listening may appear more sincere, or as put by one older adult, curiosity "…creates human connections with others often that allow understanding to develop." Expressing curiosity may thus, to older adults, signal active engagement with others and their environment, viewing that as a sign of character.

While older adults were more likely to view curiosity as a positive trait, younger adults more commonly indicated that curiosity should be expressed in moderation, highlighting an interesting age effect. Younger adults were relatively more likely to describe curiosity as a double-edged sword: It can have positive and negative consequences, depending on how it manifests. This age difference in describing curiosity's valence makes sense in light of our finding that younger adults were also much more likely to highlight curiosity as potentially harmful, if "...invading privacy…" or "being interested in the wrong topics…", for example. When we are younger, our curiosity might get us into trouble. With greater lived experience, curiosity is redirected towards expansive and joyful goals, again pointing to the potential influence of SST [15]. As time horizons narrow with age, older adults may prefer to deploy curiosity for more positive ends (e.g., learning a new recipe), which is less likely to lead to negative affect. By contrast, younger adults are more likely to deploy curiosity for both positive and negative ends (e.g., trying a drug), which is necessary for knowledge acquisition. A related construct, morbid curiosity (i.e., curiosity for negative stimuli), may also represent a dimension along which younger and older adults differ in their experience of curiosity. Accordingly, future work employing more theoretically grounded qualitative approaches may further elucidate how additional forms of curiosity—including those experienced for negative information—are experienced across the lifespan.

## Limitations

The present study has several limitations. First, both the younger and older adult samples were primarily female, and older adults were predominantly White (Table 1). Future research is needed to replicate the present study with a more ethnically and gender diverse sample of participants. Consistent with findings from a recent review, curiosity is predominantly studied in Western countries [33]. Accordingly, an important future aim is to sample from a broader range of cultures to explore

its potential effects—in addition to age—on how people conceptualize and experience curiosity. An additional limitation involves the order in which our quantitative and qualitative questions were presented to participants. The original 5DC and InC questionnaires—which empirically measure various curiosity dimensions—were presented prior to participants defining curiosity. As such, questionnaire items may have primed participants to conceptualize curiosity in ways that align with theoretical dimensions, potentially constraining the range of responses. Accordingly, we cannot rule out the possibility that exposure to these scales influenced participants' definitions of curiosity. However, approximately half of participant responses to the *Definition Question* were coded as *Miscellaneous*, indicating that despite being exposed to the scales beforehand, participant responses still contained a considerable amount of themes that deviated from those of the scales. Importantly, the goal of the present study was not to generate novel dimensions of curiosity, but rather to explore *age* differences in how individuals conceptualize curiosity relative to established theoretical frameworks. Accordingly, responses were coded deductively using existing curiosity dimensions, and any priming would be expected to operate similarly across age groups. Therefore, while priming may have influenced the overall content of responses, it is less likely to account for observed age-differences in conceptualizations. Nevertheless, future research would benefit from presenting qualitative questions prior to any standardized measures of curiosity to minimize potential priming effects. Additionally, employing more intensive qualitative methods, such as semi-structured interviews (e.g., [34]), may yield richer insights into participants' experiences. Finally, given the link between socioeconomic status (SES) and curiosity (i.e., higher SES is associated with higher levels of some curiosity dimensions; [35]), it is important for future work to include larger, more socioeconomically diverse samples, although the present samples were more or less equally distributed along the SES scale employed.

## Conclusions

While the empirical study of curiosity has been fruitful from a theoretical standpoint, little work has considered lay perspectives of curiosity. The present study took a first step in exploring this open question, employing qualitative techniques to investigate how younger and older adults define and experience curiosity. We found that along certain dimensions (e.g., deprivation sensitivity, social curiosity), younger and older adults differed in the extent to which they described curiosity, while also frequently describing the trait in ways that diverged from existing definitions (i.e., miscellaneous). Additionally, we found that younger and older adults differed in that older participants were more likely to describe curiosity as a positive trait (e.g., as it indicates one's sincerity). In contrast, younger adults more frequently described curiosity as a trait to be expressed in moderation (e.g., due to its potentially harmful consequences). The present findings align with empirical work on age differences in curiosity, suggesting that curiosity is driven more by deprivation in younger years, shifting to more interest-driven motives in older age.

Importantly, our findings touch on an open question in the aging literature: What happens to curiosity as we age? Age differences observed across state and trait measures of curiosity might be influenced by *how* younger and older adults conceptualize curiosity. By incorporating lay perspectives qualitatively, we offer a complementary conceptualization of what it means to be curious, beyond what quantitative work alone can capture. To our knowledge, the present work provides the first look at how lay conceptualizations of curiosity may vary with age—an important consideration given curiosity's role in lifelong learning and well-being [36].

## Supporting information

**S1 Table. Chi-Square Tests for Definition Question Codes.** *Note.* For all chi-square values, degrees of freedom (*df*) = 1. Under younger and older adult sample sections, numbers in parentheses indicate total percentages for corresponding code frequencies. Absence of $\chi^2$ and 95% CI values indicate that frequencies across both samples were too low for accurate comparisons. Benjamini-Hochberg (BH) corrected p-values are also included to correct for multiple comparisons. * *p* < .05. ** *p* < .01. *** *p* ≤ .001.
(DOCX)

**S2 Table. Chi-Square Tests for Valence Question Codes.** *Note.* For all chi-square values, degrees of freedom (*df*) = 1. Under younger and older adult sample sections, numbers in parentheses indicate total percentages for corresponding code frequencies. Absence of $\chi^2$ and 95% CI values indicate that frequencies across both samples were too low for accurate comparisons. Benjamini-Hochberg (BH) corrected p-values are also included to correct for multiple comparisons. * *p <.05.* ** *p <.01.* *** *p ≤.001.*
(DOCX)

**S1 Text. Category Code Combinations.**
(DOCX)

**S1 Fig. Frequency Counts for Combinations of Coding Categories in Younger Adults: Valence of Curiosity.** *Note.* Frequency counts for top (i.e., appear in ≥ 10% of all responses) combinations of initial and rationale coding category combinations are shown for the younger adult sample, illustrating how and why curiosity is experienced (i.e., in terms of its valence) as either a positive or negative trait, or a trait to be expressed in moderation.
(PDF)

**S3 Table. Most Frequent Category Code Combinations for Younger Adult Sample.** *Note.* Order of frequencies presented in order of most to least frequent.
(DOCX)

**S2 Fig. Frequency Counts for Combinations of Coding Categories in Older Adults: Valence of Curiosity.** *Note. Frequency counts for top (i.e., appear in ≥ 10% of all responses) combinations of initial and rationale coding category combinations are shown for the older adult sample, illustrating how and why curiosity is experienced (i.e., in terms of its valence) as either a positive or negative trait, or a trait to be expressed in moderation.*
(PDF)

**S4 Table. Most Frequent Category Code Combinations for Older Adult Sample.** *Note.* Order of frequencies presented in order of most to least frequent.
(DOCX)

## Author contributions

**Conceptualization:** Michelle E. Hirsch, Andrée-Ann Cyr.

**Data curation:** Michelle E. Hirsch, Andrée-Ann Cyr.

**Formal analysis:** Michelle E. Hirsch, Tarnpreet Virk, Christina Chang, Buddhika Bellana, Andrée-Ann Cyr.

**Funding acquisition:** Andrée-Ann Cyr.

**Investigation:** Michelle E. Hirsch, Tarnpreet Virk, Andrée-Ann Cyr.

**Methodology:** Michelle E. Hirsch, Tarnpreet Virk, Andrée-Ann Cyr.

**Project administration:** Michelle E. Hirsch, Andrée-Ann Cyr.

**Resources:** Andrée-Ann Cyr.

**Supervision:** Michelle E. Hirsch, Tarnpreet Virk, Buddhika Bellana, Andrée-Ann Cyr.

**Validation:** Michelle E. Hirsch, Tarnpreet Virk, Christina Chang, Andrée-Ann Cyr.

**Visualization:** Michelle E. Hirsch, Tarnpreet Virk.

**Writing – original draft:** Michelle E. Hirsch, Tarnpreet Virk, Andrée-Ann Cyr.

**Writing – review & editing:** Michelle E. Hirsch, Tarnpreet Virk, Buddhika Bellana, Andrée-Ann Cyr.

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
