## [Decision Letter · Decision Letter 0]

18 Feb 2026

Dear Dr. Hirsch,

Thank you for submitting your manuscript to PLOS ONE. After careful consideration, we feel that it has merit but does not fully meet PLOS ONE’s publication criteria as it currently stands. Therefore, we invite you to submit a revised version of the manuscript that addresses the points raised during the review process.

The manuscript has been evaluated by three reviewers, and their comments are available below.



We look forward to receiving your revised manuscript.

Kind regards,

Steve Zimmerman, PhD

Senior Editor, PLOS One

Journal Requirements:

“This work as supported by the Natural Sciences and Engineering Research Council of Canada (#RGPIN-2019-06296 to Andrée-Ann Cyr).”

3. We noted in your submission details that a portion of your manuscript may have been presented or published elsewhere. “Table 1 was included in a manuscript that is currently under review elsewhere. Table 1 includes participant demographic information and is included both here, and in Hirsch et al., 2025 because the same sample was used in both manuscripts. No shared results, data, or figures are included in pending manuscripts, thus this does not constitute dual publication.” Please clarify whether this [conference proceeding or publication] was peer-reviewed and formally published. If this work was previously peer-reviewed and published, in the cover letter please provide the reason that this work does not constitute dual publication and should be included in the current manuscript.

4. We note that you have indicated that there are restrictions to data sharing for this study. PLOS only allows data to be available upon request if there are legal or ethical restrictions on sharing data publicly. For more information on unacceptable data access restrictions, please see http://journals.plos.org/plosone/s/data-availability#loc-unacceptable-data-access-restrictions.  .  .  .

Reviewers' comments:

Reviewer's Responses to Questions

**Comments to the Author**

1. Is the manuscript technically sound, and do the data support the conclusions?

Reviewer #1: Yes

Reviewer #2: Partly

Reviewer #3: Yes

2. Has the statistical analysis been performed appropriately and rigorously?

Reviewer #1: Yes

Reviewer #2: N/A

Reviewer #3: Yes

3. Have the authors made all data underlying the findings in their manuscript fully available?

Reviewer #1: Yes

Reviewer #2: No

Reviewer #3: No

4. Is the manuscript presented in an intelligible fashion and written in standard English?

Reviewer #1: Yes

Reviewer #2: Yes

Reviewer #3: Yes

Reviewer #1: The manuscript presents a qualitative exploration of age-related differences in lay conceptualizations and valence of curiosity, using open-ended responses from younger (18-26) and older (62-94) adults.

The study innovatively bridges a gap in curiosity research by examining lay definitions against established multidimensional scales like the 5DC and InC, revealing ~50% "miscellaneous" responses that challenge theory-driven models. Age comparisons add novelty, showing older adults emphasize social aspects and positivity more than younger ones.

Strengths: Robust hybrid content analysis with high inter-rater reliability (κ ≥ 0.83 across samples), Adequate powered sample (N=155 post-exclusions) via convenience pools, with clear exclusion criteria for data quality, Transparent reporting of chi-square tests and frequencies, supported by tables/figures (e.g., joyous exploration dominant in both groups)

Weaknesses: Potential priming bias:

1. Quantitative curiosity scales administered before open-ended questions may influence responses, despite miscellaneous codes mitigating this.

2. Limited generalizability due to demographics—primarily female samples, older adults mostly White, convenience sampling from York University pools.

3. Exploratory design lacks quantitative depth; chi-squares are descriptive but not adjusted for multiple comparisons.

Key findings include age differences (e.g., older adults higher on social curiosity, younger on deprivation/thrill-seeking) and valence shifts (older: more positive/sincerity; younger: moderation due to harms like privacy invasion). ~50% miscellaneous suggests lay views exceed current frameworks.

Recommendation: Accept with minor revisions: Expand diversity in future work, clarify priming impacts, and provide supplementary code frequencies (e.g., S1/S2 Tables). Strong contribution to developmental psychology of curiosity.

Reviewer #2: I had the opportunity to review the manuscript titled “Age differences in the conceptualization and experience of curiosity: A qualitative study”. The manuscript addresses the important topic of curiosity. However, it attempts to take an approach different from standard curiosity research that is typically quantitative by employing qualitative methods.

While I appreciate the effort of the researchers, I think the study falls short of a rigorous and insightful qualitative study that would provide deep insights. The main data is essentially open ended responses to a survey that is mostly quantitative using 5DC and InC items. And a major flaw is that these items are presented first before the open ended responses, which leads to all sorts of problems as the authors acknowledge.

And more importantly, I would expect qualitative research to provide deeper insights into participants' responses via more proving prompts or follow-up interviews, for example. Instead, the respondent data is mostly quite superficial from a paid survey where the researchers had to enforce a minimum response length. Questions such as why the respondents said what they said, why they did not say certain things, how their views on curiosity may have changed over time (which is one way to address the question of curiosity differing by age), etc. are all left unaddressed.

The authors seemed to have collected quantitative data from 5DC and InC but chose not to report them here (or triangulate with the qualitative data) (perhaps, saving for another publication?).

There have been other studies that have employed qualitative approaches on curiosity (including mixed methods) with much more methodological rigor that I would have expected this manuscript to at least match. They used more probing prompts, interviews, combined quantitative data, etc. to obtain, in my opinion, deeper insights. Below are some examples:

Han, J., Way, N., Yoshikawa, H., & Clarke, C. (2025). Interpersonal curiosity and its association with social and emotional skills and well-being during adolescence. Journal of Adolescent Research, 40(3), 636-668.

Birenbaum, M., Alhija, F. N. A., Shilton, H., Kimron, H., Shahor, N., & Rosanski, R. (2023). Curious minds: Evidence from interviews with renowned experts in five curiosity-dominant fields.

Birenbaum, M., Alhija, F. N. A., Shilton, H., Kimron, H., Rosanski, R., & Shahor, N. (2019). A further look at the five-dimensional curiosity construct. Personality and Individual Differences, 149, 57-65.

I also question some of the claims made about 5DC and InC not able to capture some participant responses such as individual differences and innovation. The point of 5DC is to measure individual differences, cluster, and build curiosity profiles of individuals. And innovation partly overlaps with novelty aspects of I-type curiosity by Litman.

It’s unclear what the unit of analysis is for the code. It’s clear that some participant responses are long and cover different aspects. I assume those responses are being given multiple codes? But some examples given are very short with ellipses, so I assume they’ve been trimmed by the authors?

Minor comments:

I don’t see how refs 1 and 2 can be used to support the statement “seen a resurgence in research on human curiosity”.

Consider a more recent reference in PLoS One that can replace current ref 15: https://doi.org/10.1371/journal.pone.0320600

Reviewer #3: This study involves a qualitative analysis of curiosity definitions (what it is) and valence (is it positive or negative) for a group of younger and older adults. Overall, I find the study informative and interesting to the broader field trying to understand how to best understand and conceptualize curiosity, especially across the adult lifespan. I only have minor comments and clarifications.

1. My first question is about the fact that participants' "initial responses" to the valence question were coded. It's not entirely clear what the "initial response" is - is that the first few words that address the issue of valence? Was there a clear cutoff for initial response vs. rationale? Additionally, were there any cases where the rationale was overall different from the initial response? (e.g., they said it was positive but discussed negative aspects throughout the response)

2. It's not clear how the categories were decided on for the post-hoc coding into motivated learning, critical process, etc. Was there any specific process for coming up with these categories? And could the responses also count in more than one of these categories?

3. Similarly, it's interesting that morbid curiosity is a somewhat well-known phenomenon and these ideas tended to come up (e.g., the person who said they always ask how people died because they are interested, even if it's considered rude!). Did the authors consider including a morbid curiosity category for responses? The discussion of the potential for this category to be accounted for in measures of curiosity could also be expanded.

4. Were the responses of similar length across age groups? It seems like this was conducted on the computer, and older adults may be less accustomed to computers (though this is less of a concern as time passes). Is it possible that one age group expanded on their ideas more than another?

5. Could you say more about how older adults were recruited? I do not know what the Glendon Research Participant Pool is - are these participants involved in the university setting?

6. I think some of the references are off - especially in the 9-15 range.

.

Reviewer #1: **Yes:** Kundan Lal VermaKundan Lal VermaKundan Lal VermaKundan Lal Verma

Reviewer #2: No

Reviewer #3: No

---

## [Author Response · Author response to Decision Letter 1]

23 Feb 2026

Editor:

Thank you for providing us with the opportunity to revise our manuscript. Please find our response below.

We have updated the manuscript to meet PLOS ONE’s style requirements.

Thank you for pointing this out. We have now included the above statement in the cover letter.

3. We noted in your submission details that a portion of your manuscript may have been presented or published elsewhere. “Table 1 was included in a manuscript that is currently under review elsewhere. Table 1 includes participant demographic information and is included both here, and in Hirsch et al., 2025 because the same sample was used in both manuscripts. No shared results, data, or figures are included in pending manuscripts, thus this does not constitute dual publication.” Please clarify whether this [conference proceeding or publication] was peer-reviewed and formally published. If this work was previously peer-reviewed and published, in the cover letter please provide the reason that this work does not constitute dual publication and should be included in the current manuscript.

We are confirming that while Hirsch et al., 2026 is under review elsewhere, it has not been formally published. We have included an additional statement indicating this in the cover letter.

4. We note that you have indicated that there are restrictions to data sharing for this study. PLOS only allows data to be available upon request if there are legal or ethical restrictions on sharing data publicly. For more information on unacceptable data access restrictions, please see http://journals.plos.org/plosone/s/data-availability#loc-unacceptable-data-access-restrictions. Before we proceed with your manuscript, please address the following prompts:

If there are ethical or legal restrictions on sharing a de-identified data set, please explain them in detail (e.g., data contain potentially identifying or sensitive patient information, data are owned by a third-party organization, etc.) and who has imposed them (e.g., a Research Ethics Committee or Institutional Review Board, etc.). Please also provide contact information for a data access committee, ethics committee, or other institutional body to which data requests may be sent.

If there are no restrictions, please upload the minimal anonymized data set necessary to replicate your study findings to a stable, public repository and provide us with the relevant URLs, DOIs, or accession numbers.

We have now uploaded the minimal anonymized data set necessary to replicate our findings on osf: osf.io/vqxfr. There are no ethical or legal restrictions on sharing a de-identified data set. We do, however, note that some participants did not consent for us to share their qualitative data, and thus, their raw responses are not included in the public dataset.

Reviewer #1:

1. Weaknesses: Potential priming bias: Quantitative curiosity scales administered before open-ended questions may influence responses, despite miscellaneous codes mitigating this. Limited generalizability due to demographics—primarily female samples, older adults mostly White, convenience sampling from York University pools. Exploratory design lacks quantitative depth; chi-squares are descriptive but not adjusted for multiple comparisons.

Thank you for pointing out that chi-squares were not adjusted for multiple comparisons. We have now included all fdr-adjusted p-values for all tests conducted (S1 Table, S2 Table).

2. Recommendation: Expand diversity in future work, clarify priming impacts, and provide supplementary code frequencies (e.g., S1/S2 Tables). Strong contribution to developmental psychology of curiosity.

Thank you for these helpful recommendations. In the revised manuscript (page 35-36), we have expanded the Discussion to further emphasize the importance of diversity in future work and to clarify the potential impacts of priming. Code frequencies for younger and older samples are reported within the subheadings of each coding category. In addition, results from the S1 and S2 tables are reported in the main text where significant age differences were observed.

Reviewer #2:

1. I had the opportunity to review the manuscript titled “Age differences in the conceptualization and experience of curiosity: A qualitative study”. The manuscript addresses the important topic of curiosity. However, it attempts to take an approach different from standard curiosity research that is typically quantitative by employing qualitative methods. While I appreciate the effort of the researchers, I think the study falls short of a rigorous and insightful qualitative study that would provide deep insights. The main data is essentially open ended responses to a survey that is mostly quantitative using 5DC and InC items. And a major flaw is that these items are presented first before the open-ended responses, which leads to all sorts of problems as the authors acknowledge. And more importantly, I would expect qualitative research to provide deeper insights into participants' responses via more proving prompts or follow-up interviews, for example. Instead, the respondent data is mostly quite superficial from a paid survey where the researchers had to enforce a minimum response length. Questions such as why the respondents said what they said, why they did not say certain things, how their views on curiosity may have changed over time (which is one way to address the question of curiosity differing by age), etc. are all left unaddressed. The authors seemed to have collected quantitative data from 5DC and InC but chose not to report them here (or triangulate with the qualitative data) (perhaps, saving for another publication?). There have been other studies that have employed qualitative approaches on curiosity (including mixed methods) with much more methodological rigor that I would have expected this manuscript to at least match. They used more probing prompts, interviews, combined quantitative data, etc. to obtain, in my opinion, deeper insights. Below are some examples:

Han, J., Way, N., Yoshikawa, H., & Clarke, C. (2025). Interpersonal curiosity and its association with social and emotional skills and well-being during adolescence. Journal of Adolescent Research, 40(3), 636-668.

Birenbaum, M., Alhija, F. N. A., Shilton, H., Kimron, H., Shahor, N., & Rosanski, R. (2023). Curious minds: Evidence from interviews with renowned experts in five curiosity-dominant fields.

Birenbaum, M., Alhija, F. N. A., Shilton, H., Kimron, H., Rosanski, R., & Shahor, N. (2019). A further look at the five-dimensional curiosity construct. Personality and Individual Differences, 149, 57-65.

We thank the reviewer for their thoughtful concerns about our qualitative design. We agree that qualitative approaches that allow for probing and iterative clarification can provide richer insight into participants’ reasoning and lived experiences. In the present study, however, our goal was to conduct structured content analysis of responses to questions about lay conceptualizations of curiosity among younger and older adults, rather than in-depth phenomenological interviews (like Birenbaum et al., 2023). Our online design prioritized breadth of responses over the depth of individual conceptualizations. While this approach limits our ability to fully explore participants’ response motivations (e.g., why participants did or did not say certain things, and how their views on curiosity may have changed over time), it enables us to characterize overall patterns in how younger and older adults conceptualize curiosity using common prompts. Accordingly, our findings should be interpreted as providing an initial descriptive overview rather than an in-depth phenomenological account. To our knowledge, age differences in how people define curiosity have not been directly examined, and we position our study as a first exploratory step. We now expand on this (page 36), noting that future works employing more intensive qualitative methods may be better-suited to further understand how and why lay conceptualizations of curiosity differ as a function of age.

Additionally, as you pointed out, participants completed the 5DC and InC scale items. These measures were included to address separate research questions, which are reported elsewhere (p. 8). We agree that carryover effects from completing the 5DC and InC items represent a limitation. In the revised manuscript (pp. 35–36), we expand our discussion of this issue and clarify that the primary aim of the present study was to examine age differences in how curiosity is conceptualized, rather than to derive novel dimensions of curiosity. While completing the scale items may have influenced the overall content of responses, it is less clear that such priming would systematically differ between younger and older adults. As such, the observed differences between age groups in coding categories remain informative with respect to our primary research question, although we cannot definitively rule out the influence of priming on participants’ responses. Importantly, over half of responses were coded as miscellaneous, indicating that participants frequently articulated themes that extended beyond the dimensions captured by the 5DC and InC measures.

2. I also question some of the claims made about 5DC and InC not able to capture some participant responses such as individual differences and innovation. The point of 5DC is to measure individual differences, cluster, and build curiosity profiles of individuals. And innovation partly overlaps with novelty aspects of I-type curiosity by Litman.

Thank you for pointing this out. Indeed, the 5DC is designed to capture individual differences (in addition to clustering and building curiosity profiles), and innovation partly overlaps with novelty aspects of Litman’s I-type curiosity. That said, items from the 5DC and InC were collated to serve as code definitions guiding the deductive coding of participant responses to Question 1. Although the innovation code relates to Litman’s I-type curiosity, and individual differences are captured when participants respond directly to the 5DC scale, our coding scheme focused on mapping open-ended responses onto predefined conceptual categories. Responses that did not align with these predefined codes were therefore assigned to separate categories rather than being subsumed under 5DC-derived codes (e.g., innovation, individual differences).

3. It’s unclear what the unit of analysis is for the code. It’s clear that some participant responses are long and cover different aspects. I assume those responses are being given multiple codes? But some examples given are very short with ellipses, so I assume they’ve been trimmed by the authors?

While responses were required to have a minimum of 250 characters, response lengths varied across participants. The unit of analysis for coding was the full participant response. Accordingly, multiple codes were assigned to a single response when it reflected more than one coding category. We have now included additional clarification on how codes were assigned (pages 9-10).

For descriptive purposes, some responses were trimmed by the authors and presented as code exemplars, which may result in shorter excerpts with ellipses. We have now also provided a more comprehensive dataset (osf.io/vqxfr) containing full participant responses, where consent for data sharing was provided.

4. I don’t see how refs 1 and 2 can be used to support the statement “seen a resurgence in research on human curiosity”.

Thank you for pointing this out. You are correct that References 1 and 2 did not appropriately support the statement regarding a “resurgence in research on human curiosity.” This was an error on our part. In the revised manuscript, we have replaced these citations with more appropriate and up-to-date references that directly support this claim.

5. Consider a more recent reference in PLoS One that can replace current ref 15: https://doi.org/10.1371/journal.pone.0320600

Thank you for this recommendation. We have now replaced the current ref 15 with Whatley et al. (2025; page 4).

Reviewer #3:

1. My first question is about the fact that participants' "initial responses" to the valence question were coded. It's not entirely clear what the "initial response" is - is that the first few words that address the issue of valence? Was there a clear cutoff for initial response vs. rationale? Additionally, were there any cases where the rationale was overall different from the initial response? (e.g., they said it was positive but discussed negative aspects throughout the response)

For the valence question, the initial response was defined as the participant’s classification of curiosity as positive, negative, or something to be expressed in moderation. This represents the “what” component of the response. In contrast, rationales were defined as any part of the response that followed this initial classification, reflecting the “why” underlying the participant’s judgment.

For example, if a participant responded, “Positive because it allows me to learn more about myself,” the response would be coded as positive for the initial code category, and personal growth for the rationale code category. We have now included additional clarification on how initial and rationale codes were assigned (pages 9-10).

Because rationales were not inherently positive or negative in valence, instances of mismatch between initial classifications and rationale codes were rare.

2. It's not clear how the categories were decided on for the post-hoc coding into motivated learning, critical process, etc. Was there any specific process for coming up with these categories? And could the responses also count in more than one of these categories?

Thank you for bringing up this point of confusion. For the post-hoc coding of the valence question, the process was as follows. Briefly, two primary coders independently reviewed 50% of the responses. Following this initial review, each author began grouping responses into coding categories based on observed patterns, iteratively developing and refining category names based on the data. After completing this step, the authors met to discuss and reconcile category definitions, adjusting category names and descriptions until consensus was reached. Following this, the coding scheme was applied to the full dataset, with responses assigned to initial and rationale coding categories. Coding consistency was assessed by an additional coder on a subset of responses, with discrepancies resolved through discussion.

Responses could be assigned to more than one category when they reflected multiple themes. We have now clarified both the coding procedure and the use of multiple coding in the manuscript (pages 9-10).

3. Similarly, it's interesting that morbid curiosity is a somewhat well-known phenomenon and these ideas tended to come up (e.g., the person who said they always ask how people died because they are interested, even if it's considered rude!). Did the authors consider including a morbid curiosity category for responses? The discussion of the potential for this category to be accounted for in measures of curiosity could also be expanded.

This is a very interesting point. As the valence question was coded inductively (i.e., post-hoc), coding categories were derived directly from participants’ responses rather than being guided by existing theoretical frameworks (see response to above comment). Accordingly, we did not include a predefined morbid curiosity category during the coding process.

That said, we agree that some responses—such as those reflecting interest in death or other negatively valenced content—a

---

## [Decision Letter · Decision Letter 1]

13 Apr 2026

Age differences in the conceptualization and experience of curiosity: A qualitative study

PONE-D-25-66736R1

Dear Dr. Hirsch,

We’re pleased to inform you that your manuscript has been judged scientifically suitable for publication and will be formally accepted for publication once it meets all outstanding technical requirements.

Kind regards,

Vanessa Carels

Staff Editor

PLOS One

Additional Editor Comments (optional):

Reviewers' comments:

Reviewer's Responses to Questions

**Comments to the Author**

Reviewer #1: All comments have been addressed

Reviewer #2: All comments have been addressed

Reviewer #3: All comments have been addressed

2. Is the manuscript technically sound, and do the data support the conclusions?

Reviewer #1: Yes

Reviewer #2: Partly

Reviewer #3: Yes

3. Has the statistical analysis been performed appropriately and rigorously?

Reviewer #1: Yes

Reviewer #2: Yes

Reviewer #3: Yes

4. Have the authors made all data underlying the findings in their manuscript fully available?

Reviewer #1: Yes

Reviewer #2: (No Response)

Reviewer #3: Yes

5. Is the manuscript presented in an intelligible fashion and written in standard English?

Reviewer #1: Yes

Reviewer #2: Yes

Reviewer #3: Yes

Reviewer #1: Dear Author(s)

I have carefully reviewed the revised manuscript titled “Age differences in the conceptualization and experience of curiosity: A qualitative study” (PONE-D-25-66736R1). The authors have thoroughly addressed the concerns raised in the previous review round, and the manuscript has been substantially strengthened as a result.

Major Strengths

Clear Contribution to the Literature

The manuscript provides a novel and meaningful contribution to the curiosity literature by examining age differences in lay conceptualizations of curiosity. While much prior work has relied on theory-driven quantitative scales (e.g., the 5DC and InC), this study offers an exploratory qualitative perspective across the lifespan. This fills a clear empirical gap.

Methodological Transparency and Rigor

The authors have:

Clarified the hybrid deductive–inductive content analysis approach.

Clearly defined the unit of analysis and coding procedures.

Provided strong inter-rater reliability statistics.

Included FDR-adjusted p-values for chi-square analyses.

Expanded discussion of limitations (e.g., priming effects, sampling constraints).

These revisions significantly improve methodological clarity and credibility.

Thoughtful Engagement with Reviewer Concerns

The authors have constructively responded to critiques regarding:

Priming effects from quantitative scales.

Depth versus breadth in qualitative methodology.

Coding development and rationale classification.

Data transparency (OSF upload).

Reference updates and citation corrections.

The discussion now appropriately frames the study as an exploratory, descriptive first step rather than a deep phenomenological investigation.

Data Transparency and Ethics

The data availability statement, OSF repository upload, ethics approval (York University Office of Research Ethics, certificate #2024-294), and funding disclosure (Natural Sciences and Engineering Research Council of Canada) are clearly reported and compliant with journal standards.

Interpretation and Framing

Importantly, the manuscript now more accurately positions its findings:

As descriptive age-related differences in conceptualization rather than causal or developmental claims.

As complementary to established frameworks (e.g., 5DC), rather than as a critique of them.

As groundwork for future in-depth qualitative or mixed-method research.

The acknowledgment of limitations (e.g., convenience sampling, limited diversity, potential priming, non-interview format) is appropriate and balanced.

Minor Observations

The manuscript appears ready for publication. Any remaining minor stylistic or formatting adjustments can be handled during copyediting.

Reviewer #2: (No Response)

Reviewer #3: The authors have addressed my comments, and I have no further suggestions. I appreciate the opportunity to review this manuscript.

.

Reviewer #1: **Yes:** Kundan Lal VermaKundan Lal VermaKundan Lal VermaKundan Lal Verma

Reviewer #2: No

Reviewer #3: No

---

## [Editor Report · Acceptance letter]

PONE-D-25-66736R1

PLOS One

Dear Dr. Hirsch,

I'm pleased to inform you that your manuscript has been deemed suitable for publication in PLOS One. Congratulations! Your manuscript is now being handed over to our production team.

Kind regards,

on behalf of

Dr. Vanessa Carels

Staff Editor

PLOS One